# Structural Characterization and In Vitro Anti-Inflammatory Activity of Polysaccharides Isolated from the Fruits of *Rosa laevigata*

**DOI:** 10.3390/ijms25042133

**Published:** 2024-02-09

**Authors:** Song Peng, Pengfei Gu, Ningning Mao, Lin Yu, Tianyu Zhu, Jin He, Yang Yang, Zhenguang Liu, Deyun Wang

**Affiliations:** 1Institute of Traditional Chinese Veterinary Medicine, College of Veterinary Medicine, Nanjing Agricultural University, Nanjing 210095, China; payson@stu.njau.edu.cn (S.P.); 2021207065@stu.njau.edu.cn (N.M.); yulin@stu.njau.edu.cn (L.Y.); 2020207036@stu.njau.edu.cn (T.Z.); 2021207042@stu.njau.edu.cn (J.H.); yangyang@njau.edu.cn (Y.Y.); 2021080@njau.edu.cn (Z.L.); 2MOE Joint International Research Laboratory of Animal Health and Food Safety, College of Veterinary Medicine, Nanjing Agricultural University, Nanjing 210095, China; 3College of Traditional Chinese Veterinary Medicine, Hebei Agricultural University, Baoding 071001, China; pfgu@hebau.edu.cn

**Keywords:** polysaccharide, structure characterization, M1 macrophage polarization

## Abstract

RLPa-2 (Mw 15.6 kDa) is a polysaccharide isolated from *Rosa laevigata Michx*. It consists of arabinose (Ara), galactose (Gal), rhamnose (Rha), glucose (Glc), xylose (Xyl), and galacturonic acid (Gal-UA) with a molar ratio of 1.00:0.91:0.39:0.34:0.25:0.20. Structural characterization was performed by methylation and NMR analysis, which indicated that RLPa-2 might comprise →6)-α-D-Galp-(1→, →4)-α-D-GalpA-(1→, α-L-Araf-(1→, →2,4)-α-D-Glcp-(1→, β-D-Xylp, and α-L-Rhap. In addition, the bioactivity of RLPa-2 was assessed through an in vitro macrophage polarization assay. Compared to positive controls, there was a significant decrease in the expression of M1 macrophage markers (CD80, CD86) and p-STAT3/STAT3 protein. Additionally, there was a down-regulation in the production of pro-inflammatory mediators (NO, IL-6, TNF-α), indicating that M1 macrophage polarization induced with lipopolysaccharide (LPS) and interferon-γ (IFN-γ) stimulation could be inhibited by RLPa-2. These findings demonstrate that the RLPa-2 might be considered as a potential anti-inflammatory drug to reduce inflammation.

## 1. Introduction

*Rosa laevigata Michx*, the dried mature pyriform fruit of the *Rosaceae* plant, is widely used as a medicinal and edible plant in China [1]. *R. laevigata* is recognized as a health food resource by the Chinese Ministry of Health, and served as a medicine for the spleen and for relieving diarrhea in Traditional Chinese Medicine (TCM) approximately 1000 years ago. Pharmacological studies have demonstrated that *R. laevigata* possesses excellent medicinal value, which is characterized by its antioxidant, hypolipidemic, and anti-inflammatory effects [2]. Its primary components include polysaccharides, triterpenoid acids, vitamin C, steroids, polyphenols, and saponins. Extracts from *R. laevigata* have been reported to effectively alleviate oxidative stress, apoptosis, and inflammatory reactions [3,4]. Moreover, polysaccharides that are crucial components of the *R. laevigata* have increasingly become the focus of research due to their biological activity. Zhan et al. [5] demonstrated that a novel acidic polysaccharide isolated from the fruits of *R. laevigata* could recognize pattern recognition receptors (PRRs) of macrophages and enhance immunomodulatory activity by activating MAPKs and NF-κB signaling pathways. Zhang et al. [6] revealed that a neutral polysaccharide (RLP50-2) purified from the fruits of *R. laevigata* was mainly composed of →4)-α-D-Glcp-(1→ and →6)-α-D-Glcp-(1→. Meanwhile, RLP50-2 also exhibited significant antitumor activities. Liu et al. [7] isolated two novel homogeneous selenium-containing polysaccharides from *R. laevigata Michx* fruits (Se-RLFPs), which could modulate oxidative stress by activating the Nrf2/HO-1 signaling pathway. However, the effect of polysaccharides isolated from the fruits of *R. laevigata* on inflammation remains unclear.

Plant polysaccharides, naturally bioactive macromolecules, have attracted attention for their excellent bioactivity and safety [8]. It has been reported that polysaccharides possess anti-inflammatory activity. For example, a polysaccharide isolated from *Saccharina japonica* reduced nitric oxide (NO) production and down-regulated the expression of MAPK and NF-κB signaling pathways. The results indicated a potential correlation with its main chain being composed of 1, 6-Galp [9]. *Houttuynia cordata* polysaccharide could significantly inhibit NO secretion in inflammatory cells, also inhibiting the activation of the NF-κB pathway to perform anti-inflammatory effects by reducing mRNA expression of intracellular inducible factors (iNOS, TNF-α, and IL-1β). This may be attributed to its main chain composed of →2)-α-L-Rhap-(1→, →4)-α-D-GalpA-(1→ and →4)-β-D-Galp-(1→ [10]. However, *R. laevigata* polysaccharides might have a similar main chain, making it meaningful to investigate the potential association between the structures and their anti-inflammatory effects.

Macrophages play a crucial role in the inflammation process, being one of the most dominant and widespread inflammatory cells. When stimulated by LPS and IFN-γ, macrophages polarize to the M1 phenotype, releasing inducible NO, interleukins 6 (IL-6), and tumor necrosis factor (TNF-α), with high expression of the differentiated clusters CD80, CD86, and other pro-inflammatory cytokines that contribute to the clearance of microorganisms. However, the overproduction of pro-inflammatory factors may lead to host injury. The signal transducer and activator of the transcription (STAT) pathway play an essential role in the inflammatory response. Signal transducer and activator of transcription 3 (STAT3) is a signal transducer and activator of transcription at cytokine and growth factor receptors [11,12]. It has been reported that the expression levels of phosphorylated STAT3 (p-STAT3)/STAT3 and pro-inflammatory mediators were increased in macrophages after stimulation with LPS and IFN-γ. This suggests that activated STAT3 may be involved in macrophage polarization [13]. However, there is no report about the anti-inflammatory activity of *R. laevigata* polysaccharides through the modulation of M1 macrophage polarization.

The primary objective of this study was to elucidate the structural characteristics of *R. laevigata* polysaccharide and their anti-inflammatory effects within the classical activation model of M1 macrophages induced by LPS and IFN-γ to explore associated mechanisms. This research was designed to contribute more comprehensive information to the database on the structural properties of *R. laevigata* polysaccharides and facilitate their broad application in functional foods and pharmaceuticals.

## 2. Results and Discussion

### 2.1. Isolation, Purification, and Molecular Weight Analysis of RLPa-2

The flowchart illustrating the extraction and purification of polysaccharides from *R. laevigata* is presented in Figure 1. In summary, the process began with obtaining crude polysaccharide from *R. laevigata*. Following hot water extraction, ethanol precipitation, and deproteinization using the Sevag reagent, the *R. laevigata* crude polysaccharide (RLP) was obtained. The crude polysaccharide was then subjected to purification by DEAE Sepharose Fast Flow, resulting in three fractions: RLP-1, RLP-2, and RLP-3 (Figure 2A). The RLP-2 fraction underwent further purification by collecting fractions using Sephacryl S-400HR due to its high content yielding 1.38%. Additionally, during the analysis, two spikes (RLPa-2 and RLPb-2) appeared (Figure 2B), and the purity of RLPa-2 was determined to be 90.8% using an anthrone-sulfuric acid method. Consequently, these fractions were collected for further study.

### 2.2. Molecular Weight Analysis of RLPa-2

The relative molecular mass parameters of RLPa-2 were determined by gel chromatography combined with differential angle laser scattering systems (Figure 3A,B). The molar masses ranged from 1.0 × 10^4^ to 1.0 × 10^5^ with symmetric signal peaks, indicating that RLPa-2 exhibits homogeneous relative molecular weight distribution. The conformation of polysaccharides in the solution exhibited variability. The average slope of the conformational diagram was determined to be 0.16 ± 0.1, as shown in Figure 3B. It encompasses slopes of 0.55 ± 0.09 (Mw > 1.0 × 10^4^) and −1.14 ± 0.16 (Mw ≤ 1.0 × 10^4^), as depicted in Appendix A. These findings suggested that the conformation of RLPa-2 transitions from irregular coils to more compact spheres with increasing molecular weight. Monodisperse polymers are characterized by an Mw/Mn of 1, whereas polymers with a broad molecular weight distribution exhibit an Mw/Mn significantly greater than 1 [14]. Accounting for the uncertainty of sample, the molecular weight parameters of RLPa-2 are detailed in Appendix A, where Mw and Mw/Mn are approximately 15.6 kDa and 1.7, respectively. These values indicate that it is a polymer with moderate dispersion.

### 2.3. Monosaccharide Composition Analysis

The composition of monosaccharides plays a significant role in the characterization of polysaccharides. The composition of monosaccharides was analyzed by HPAEC, as illustrated in Appendix A. RLPa-2 primarily comprises arabinose (Ara), galactose (Gal), rhamnose (Rha), glucose (Glc), xylose (Xyl), and galacturonic acid (Gal-UA) with a molar ratio of 1.00:0.91:0.39:0.34:0.25:0.20, indicating that RLPa-2 was a heteropolysaccharide. This result differs from previous reports, possibly due to differences in polysaccharide sources and extraction methods.

### 2.4. Methylation Analysis of RLPa-2

The structure information of RLPa-2 was further determined through methylation and GC-MS analysis of glycosidic linkages. The total ion chromatogram and tandem mass spectra are present in Appendix A. Glycosyl linkage patterns, molar ratios, and mass fragmentation information are shown in Table 1. Based on monosaccharide composition analysis, it was determined that RLPa-2 contained Gal-UA at 5%. To obtain accurate structural information about the RLPa-2, uronic acids were evenly divided into two parts. One part was treated with 1 mL NaBH4, and the other part was treated with 1 mL NaBD4 before methylation. This approach, with reduced polysaccharide, minimizes the generation of excess byproducts during the derivatization process, allowing more precise discrimination of uronic acids and determination of their proportions by utilizing the isotope mass difference. This ensures the accuracy of the detection results [15]. The results showed eight glycosidic bonds in RLPa-2, primarily including t-Araf-(→1, →4)-GalpA-(1→, →2,4)-Glcp-(1→, →4,6)-Galp-(1→ and →3,6)-Galp-(1→, which accounts for over 70% of the composition. In addition, small amounts of →6)-Galp-(1→, t-Rhap-(→1 and t-Xylp-(→1 were also detected. This demonstrated that RLPa-2 is a highly branched acidic heteropolysaccharide aligned with the monosaccharide composition results. However, it is important to note that the detection methods for monosaccharide composition involve quantitative measurements using external standards to calculate the molar ratios of each monosaccharide, while methylation detection relies on calculating the peak areas in GC-MS, which approximates the molar ratios of glycosidic linkages. As a result, the relative molar masses of Rha and Ara obtained through methylation exhibit slight variations from the monosaccharide composition results [16]. It has been reported that galactose in polysaccharides might be associated with anti-inflammatory effects [17], and a significant amount of galactose was found in RLPa-2.

### 2.5. NMR Analysis of RLPa-2

The structure of RLPa-2 was further elucidated through one-dimensional nuclear magnetic resonance (1D-NMR) and two-dimensional nuclear magnetic resonance (2D-NMR) techniques. NMR spectroscopy played a crucial role in determining the type of glycosyl residues, substitution site, and sequence. ^1^H NMR was used to interpret the conformation of the glycosidic bonds. In the ^1^H NMR spectrum (Figure 4A), the signals of RLPa-2 in ^1^H NMR ranged from 3 ppm to 6 ppm. The type of β-configuration predominantly appeared in the δ H 4.3–4.8 ppm region, while the α-configuration was typically observed over the 4.8 ppm region [18]. The 1H NMR spectrum displayed six anomeric proton signals with chemical shifts of δ 4.82 ppm, δ 4.98 ppm, δ 4.51 ppm, δ 5.00 ppm, δ 4.97 ppm, and δ 4.43 ppm, assigned to H-1 of A, B, C, D, E, and F. Combined with the crosspeaks observed in the HSQC spectrum (Figure 4E), six sugar residues could be assigned with their respective anomeric proton signals at δ 4.82/100.06 ppm(H1/C1), δ 4.98/98.94 ppm(H1/C1), δ 4.51/96.22 ppm(H1/C1), δ 5.00/107.52 ppm(H1/C1), δ 4.97/99.47 ppm(H1/C1), and δ 4.43/102.8 ppm(H1/C1). It is suggested that sugar residue A was probably α-D-Galp based on the anomeric proton signal at δ 4.82/100.06 ppm. As shown in Figure 4C, the H2 signal of residue A was confirmed at 3.51 ppm using the crosspeaks identified in the ^1^H-^1^H COSY spectrum (δ 4.82/3.65 ppm). Adopting the same approach, we could assign the H3 to H6 signals sequentially: δ 4.01 ppm, δ 3.82 ppm, δ 3.97 ppm, δ 3.92 ppm (H6a), and δ 3.66 ppm (H6b). Subsequently, the chemical shifts for C2 to C6 were determined as follows: δ 68.33 ppm, δ 70.04 ppm, δ 69.55 ppm, δ 74.16 ppm, and δ 68.89 ppm. Notably, the chemical shift toward the downfield of C1 and C6 suggests that this residue has been substituted at the O-1 and O-6 positions in the sugar ring. It can be inferred that the sugar residue A was likely to be →6)-α-D-Galp-(1→, which was consistent with previous reports [19]. The chemical shift signal of the polysaccharide had a broader distribution in ^13^C NMR compared to ^1^H NMR, which was distributed in the region of 90–110 ppm. According to the ^13^C NMR and HSQC spectrum (Figure 4B,E), anomalous carbon signatures of sugar residues were found, with the main chemical shifts at δ 100.06 ppm, δ 98.94 ppm, δ 96.22 ppm, δ 107.52 ppm, δ 99.47 ppm, and δ 102.8 ppm. Thus, the residues were inferred to be part of a glycosidic bond →4)-α-D-GalpA-(1→(B), →4,6)-β-D-Galp-(1→(C), α-L-Araf-(1→(D), →2,4)-α-D-Glcp-(1→(E), →3,6)-β-D-Galp-(1→(F). Although the aldehyde acid of →4)-α-D-GalpA-(1→ exists as a methoxy ester group (-COOCH3), the signal peak near δ3.62/59.8 ppm originated from the carbon signal of O-CH3 [20,21]. However, unlike the weak crosspeak signals of certain sugar residues in the COSY spectrum, the H1 signal of residue D at δ 5.00 ppm was not identified. Nevertheless, a crosspeak at δ 5.00/4.05 ppm was detected in the TCOSY spectrum (Appendix A), resulting in the assignment of δ 4.05 ppm as the H2 signal of residue D. In addition, the assignment of H and C chemical shifts for all sugar residues was completed using the TCOSY and COSY spectrum (Table 2).

The linkage relationships between sugar residues were deduced from the crosspeak signals in the HMBC and NOESY spectra. The HMBC spectrum (Figure 4F) displayed crosspeaks, correlating sugar residues B-H1 and A-C6 with a chemical shift of δ 4.98/68.89 ppm and D-H1 and E-C4 with δ 5.00/79.07 ppm. The sequence of residues in polysaccharides could be validated by the NOESY spectrum, as shown in Figure 4D. There was a crosspeak at δ 4.82/3.91 ppm between residue A-H1 and C-H6, a crosspeak at δ 4.51/3.67 ppm between residue C-H1 and F-H6, a crosspeak at δ 5/4.49 ppm between residue D-H1 and E-H4, and a cross peak at δ 4.97/3.92 ppm between residue E-H1 and A-H6.

The methylation and NMR analysis showed that the backbone chain of RLPa-2 is composed primarily of →6)-α-D-Galp-(1→, which partially contained →4)-α-D-GalpA-(1→. The side chain consists primarily in α-L-Araf-(1→ linked to the O-2 position in →2,4)-α-D-Glcp-(1→, which probably included part of β-D-Xylp or α-L-Rhap (unrecognized intact signal). In summary, the possible structure of the acidic polysaccharide RLPa-2 is depicted below (Figure 4G), where the R part of the side chain might be β-D-Xylp or α-L-Rhap.

### 2.6. Modulation of Inflammation by RLPa-2 In Vitro

#### 2.6.1. The Effect of RLPa-2 on RAW264.7 Macrophage Survival

Before assessing the in vitro anti-inflammatory activity of RLPa-2, the effect of different concentrations of RLPa-2 on macrophage cell viability was determined by MTT. As illustrated in Figure 5A, RLPa-2 (5, 10, 15, 25, 50, 100, 200, 400 μg/mL) exhibited no cytotoxicity to RAW264.7 cells and significantly promoted cell proliferation in the 50–200 μg/mL. However, at higher concentrations of RLPa-2 (400 μg/mL), the osmotic pressure of the cell culture medium was altered by the high concentrations, leading to a decline in the proliferative activity of the cells. Based on the above results, the concentrations 50 µg/mL (RLPa-2-L), 100 µg/mL (RLPa-2-M), and 200 µg/mL (RLPa-2-H) were selected for the subsequent study.

#### 2.6.2. Effect of RLPa-2 on NO Secretion and Pro-Inflammatory Cytokines of Cells

Macrophages play an essential role in inflammatory response. In response to inflammation in the body, macrophages can differentiate into pro-inflammatory macrophages (M1) that secrete the pro-inflammatory mediator NO and those that secrete the pro-inflammatory cytokines IL-6 and TNF-α to promote an inflammatory response. Numerous studies have shown that an overabundance of NO might induce a systemic inflammatory response [22]. TNF-α serves as a mediator of inflammatory dysfunction, leading to pathogenic alterations (apoptosis, and inflammation) [23]. IL-6 is a soluble mediator with multiple effects that can exert pro-inflammatory or anti-inflammatory effects and is associated with various conditions, including obesity, exercise, arthritis, and colitis [24].

Therefore, the effect of RLPa-2 on M1 macrophage inflammatory factors and mediators was assessed. As shown in Figure 5B–D, when RAW264.7 cells were stimulated with LPS and IFN-γ, the levels of NO, TNF-α, and IL-6 were significantly higher than those of the control group. In contrast, the administration of RLPa-2 (50, 100, 200 µg/mL) led to a significant decrease in the release of NO, TNF-α, and IL-6, with the most significant reduction observed in the RLPa-2-H group. These results suggest that RLPa-2 could down-regulate pro-inflammatory factors and mediators secreted by M1 macrophages in a dose-dependent manner.

#### 2.6.3. Effect of RLPa-2 on Morphological Change of M1 Macrophages

Macrophages exhibit sensitivity to environmental conditions, particularly diverse stimulation signals. Macrophages (M0 or Mφ) can undergo polarization to M1 or M2 states, each displaying distinct cellular morphology [25]. In this study, RAW264.7 cells were activated to transform into M1 phenotype using LPS and IFN-γ. Microscopic examination (Figure 6A) revealed that normal macrophages had an oval and well defined shape, with rounded cell bodies and standard forms. In contrast, M1 macrophages had numerous “tentacles” with lysosomal granules and vesicles in the cytoplasm. When co-cultured with RLPa-2, M1 macrophages resulted in significant contraction with fewer abnormal cells.

#### 2.6.4. Effect of RLPa-2 on Surface Markers of M1 Macrophages

M1 macrophages, being pro-inflammatory cells, are characterized by major histocompatibility complex (MHC) class II expression and by high levels of CD80 and CD86. In this study, the treatment group received RLPa-2 solution, while negative and positive controls were treated with DMEM solution (containing equal amounts) and LPS+IFN-γ, respectively. The effect of RLPa-2 on CD86 and CD80 expression was assessed by flow cytometry. As shown in Figure 6B, the expression of specific surface phenotypic markers of CD80 and CD86 was significantly higher in activated M1 macrophages than in negative controls. However, RLPa-2 (50, 100, and 200 µg/mL) markedly reversed this effect, with the most pronounced effect observed at 200 µg/mL. This implies that the surface markers of M1 macrophages that were induced with LPS and IFN-γ could be inhibited by RLPa-2. Previous reports have suggested that polysaccharides can impede the excessive conversion of macrophages to the M1 phenotype, aligning with the results of the present study [26,27].

#### 2.6.5. Effect of RLPa-2 on STAT3 Protein Expression in M1 Macrophages

In addition to direct anti-inflammatory effects, polysaccharides may also activate signaling pathways to prolong intracellular defense responses. It has been reported that certain plant polysaccharides participate in the STAT3 pathway to inhibit macrophage activation. While numerous signaling pathways are involved in regulating macrophage polarization, the activation of STATs is indispensable. STAT3, a member of the STAT family, plays a critical role in inflammation and tumorigenesis [28]. Therefore, Western blot was used to detect STAT3 and p-STAT3 protein levels (Figure 7). The protein bands were lighter in the control group and darker in the model group (LPS+IFN-γ), suggesting significant expression of STAT3 and p-STAT3 in M1 macrophages. Similarly, compared to the model group, protein bands were lighter and thinner with RLPa-2 treatment, significantly reducing STAT3 and p-STAT3 protein expression in macrophages with LPS and IFN-γ. Based on these results, we hypothesized that RLPa-2 exhibits anti-inflammatory capacity in vitro via the STAT3 pathway.

## 3. Materials and Methods

### 3.1. Materials

*R. laevigata* was purchased from Kanghao Herb Slicing Co., Ltd. (Bozhou, China). DEAE Sepharose Fast Flow were supplied by Sunresin New Materials Technology Co., Ltd. (Xian, China). Sephacryl S-400HR was supplied by GE Healthcare (Chicago, IL, USA). IFN-γ was provided by Pepro Tech Inc. (Rocky Hill, NJ, USA). Lipopolysaccharide (LPS) was purchased from Sigma (St Louis, MO, USA). TNF-α and IL-6 cytokine ELISA kits were purchased from MultiSciences Biotech Co., Ltd. (Nanjing, China). FITC-labeled anti-CD80 and PE-labeled anti-CD86 were purchased from eBioscience (California, CA, USA). Monosaccharide standards and dimethyl sulfoxide (DMSO) were supplied by Sigma-Aldrich Chemical Co. (St. Louis, MO, USA). All other reagents were analytical-grade.

### 3.2. Extraction and Purification of Polysaccharides from the Fruits of R. laevigata

The extraction of polysaccharides from *R. laevigata* was performed according to the previously reported method with slight modifications [29]. Simply, *R. laevigata* were extracted three times with distilled water (90 °C) in a ratio of 1:20 (*w*/*v*) for 3 h each. The aqueous extract was concentrated to one-twentieth of the total volume, then the supernatant was collected by centrifugation. The concentrate was settled with 95% ethanol to a concentration of 65% and maintained for 24 h. The sediment was collected after centrifugation, re-dissolve in water, and concentrated at 60 °C. Meanwhile, by employing the Sevag method, the concentrates were deproteinated (chloroform: butanol, *v*/*v* = 1:4) and procedure was repeated 8 times. Subsequently, the protein-free solution was dialyzed in a filter bag (cut-off Mw: 3500 Da) in flowing deionized water for 48 h to obtain crude polysaccharides (RLP) through freeze-drying. The crude polysaccharide solution (10 mg/mL) was centrifuged and filtered (0.45 µm), and loaded onto a DEAE Sepharose Fast Flow column (2.6 cm × 40 cm) pre-equilibrated with water and gradually eluted with distilled water, 0.1, 0.2 and 0.3 M NaCl solutions. Fractions containing polysaccharides (15 mL/tube) were collected for determination by the phenol-sulfuric acid method at 490 nm. Furthermore, three carbohydrate-positive fractions, RLP-1, RLP-2, and RLP-3, were compiled separately. The main fraction RLP-2 (15 mg/mL) was further purified on a Sephacryl S-400HR column (2.6 cm × 100 cm) which was eluted with distilled water (0.2 mL/min) and lyophilized to obtain the purified polysaccharide called RLPa-2.

### 3.3. Molecular Weight Measurement

Polysaccharide molecular weight was determined using the method reported previously [30]. The homogeneity and molecular weight of various fractions were measured using SEC-MALLS-RI equipped with multiple detectors, including a DAWN HELEOS-II laser photometer (Wyatt Technology Co., Santa Barbara, CA, USA), a differential refractive index detector Optilab T-rEX (Wyatt Technology Co., Santa Barbara, CA, USA), and a gel exclusion column (OH-pak SB-805 HQ, OH-pak SB-804 HQ, and OH-pak HQ 803 (300 × 8 mm)). The purified RLPa-2 (1 mg/mL) was solubilized in NaNO_3_ (0.1 M) aqueous solution with 0.02% NaN_3_ and filtered through a filter (0.45 µm). The column was kept at a consistent temperature (45 °C), with an injection volume of 100 μL and a flow rate set at 0.5 mL/min. An isocratic elution was then conducted for 100 min. The incremental specific refractive index (dn/dc) values were determined to be 0.141 mL/g in NaNO_3_ aqueous solution. Data were analyzed using ASTRA 6.1 (Wyatt Technology Co., Santa Barbara, CA, USA).

### 3.4. Analyses of Monosaccharide Composition

The samples were analyzed for their monosaccharide composition following the reported method [31]. Approximately 5 mg of the sample was hydrolyzed in the chromatography vial with 2 M trifluoroacetic acid (TFA) at 121 °C for 2 h and dried in nitrogen. Methanol was added and washed 3 times to dry. The residues were redissolved in deionized water and filtered through a filtering membrane (0.22 µm). Next, the samples were collected for determination by high performance anion exchange chromatography (HPAEC) using Dionex™ CarboPac™ PA20 liquid chromatographic column (150 × 3.0 mm, 10 µm). The HPAEC conditions are described in Appendix A. Data were acquired on the ICS5000 (Thermo Fisher Scientific, Waltham, MA, USA) and processed using chromeleon 7.2 CDS (Thermo Fisher Scientific, Waltham, MA, USA).

### 3.5. Methylation Analysis

The methylation product of RLPa-2 was prepared according to the previously reported method [15]. The methylation method was used to reduce the uronic acid content of polysaccharides. The samples were individually reduced using NaBH4 and NaBD4, followed by methylation with CH_3_I in DMSO/NaOH. The completely methylated polysaccharides were hydrolyzed, reduced, and acetylated to obtain products which were analyzed using gas chromatography-mass spectrometry (GC-MS). The detailed experimental procedures are listed in Appendix A.

### 3.6. Nuclear Magnetic Resonance (NMR) Analysis

The RLPa-2 sample (20 mg) was solubilized in 0.5 mL of D_2_O in an NMR tube, and the NMR spectra of RLPa-2 were recorded with a Bruker AVANCE NEO 500M spectrometer (Bruker, Rheinstetten, Germany) at 500 MHz. This included one-dimensional (^1^H and ^13^C) and two-dimensional NMR spectra (HMBC, ^1^H-^1^H COSY, TCOSY, and HSQC).

### 3.7. Anti-Inflammatory Activity Test In Vitro

#### 3.7.1. Cell Culture and Treatments

The murine macrophage RAW264.7 cell line was obtained from American Type Culture Bank (Shanghai, China) and incubated in Dulbecco’s modified Eagle’s medium (DMEM; Sigma, USA) supplemented with 10% heat-inactivated fetal bovine serum (FBS; Gbico, Grand Island, NY, USA) and 1% penicillin and streptomycin solution in a humidified incubator at 37 °C with 5% CO_2_.

#### 3.7.2. Cell Viability Assessment

Macrophage viability was determined using the 3-(4,5-dimethylthiazol-2-yl)-2,5-diphenyltetrazolium bromide (MTT) method, as described in our previous article. Briefly, RAW264.7 macrophages were diluted to 5 × 10^5^ cells/mL with complete medium, inoculated with 100 µL per well into 96-well plates, and incubated at 37 °C and 5% CO_2_ for 12 h. Different concentrations of RLPa-2 (5, 10, 15, 25, 50, 100, 200, 400 µg/mL) were added to the treatment group, and an equal volume of DMEM was added. All cells were incubated in an incubator for 24 h. MTT solution (5 mg/mL) was added and incubated for 4 h. Finally, the supernatant was removed to add DMSO, and the absorbance was measured at 570 nm.

#### 3.7.3. NO and Cytokine Measurement

RAW264.7 macrophages were diluted to 5 × 10^5^ cells/mL with complete medium and plated onto 96-well plates for an overnight incubation in an incubator. Subsequently, various concentrations of RLPa-2 (50, 100, 200 µg/mL) were added to coculture for 2 h. Finally, the M1 macrophage model was established by administering LPS (100 ng/mL) and IFN-γ (20 ng/mL) and cultured for 24 h. DMEM complete medium served as a blank control, while LPS and IFN-γ were employed as positive controls. Following this, cell supernatants were collected, and NO levels in the supernatants were measured by the Griess method. TNF-α and IL-6 levels were measured using multifunctional microplates following the instructions provided in the enzyme linked immunosorbent assay (ELISA) kit instructions.

#### 3.7.4. Flow Cytometric Assessment of Surface Markers of M1 Macrophages

Macrophages were diluted and plated in 24-well plates and incubated for 24 h with RLPa-2, LPS, and IFN-γ as described in Section 3.7.3. Cells were collected with a cell scraper, washed with phosphate buffered saline (PBS), stained with anti-mouse CD80-FITC (1 µg/mL) and anti-mouse CD86-PE (1 µg/mL) antibodies (eBioscience Inc., San Diego, CA, USA) for 30 min at 4 °C, protected from light, and washed twice again with PBS. The expression of CD80 and CD86 on the cell surface was measured by flow cytometry analysis (BD Accuri^®^C6 flow cytometer, San Jose, CA, USA).

#### 3.7.5. Morphological Analysis of Activated Macrophages

M1 macrophage were established by plating RAW264.7 macrophages in 6-well plates. Next, RLPa-2 was added and cocultured for 24 h followed by three washes with PBS. Macrophage morphology was observed using electron microscopy.

#### 3.7.6. Western Blot Analysis

Macrophages were cultured and stimulated as described above. Briefly, macrophage dilution suspensions were inoculated into 6-well plates. After overnight incubation, RLPa-2 (200 µg/mL) was added as outlined in Section 3.7.3. The same volume of DMEM was added to the blank control and positive control groups. Subsequently, LPS and IFN-γ were added to the sample and positive control groups and incubated for 24 h. Next, cells were washed with PBS and radioimmunoprecipitation assay buffer (RIPA) lysate containing 1% Phenylmethylsulfonyl fluoride (PMSF) was added, incubated at 4 °C, and sonicated. The supernatant was then centrifuged (10,000 rpm, 10 min) to determine the protein concentration. Western blot analysis was conducted using our previously reported method [32]. Equal proteins were subjected to sodium dodecyl sulfate-polyacrylamide gel electrophoresis (SDS-PAGE) and transferred to polyvinylidene fluoride (PVDF) membranes. Blocking was performed with 5% skim milk in PBS (0.05% Tween-20). The membranes were incubated with primary antibodies against p-STAT3 and STAT3 (Cell Signaling Technologies, Danvers, MA, USA) (4 °C, 12 h), followed by conjugation of secondary antibodies detected using ECL reagents.

### 3.8. Statistical Analysis

All experiments were performed at least in triplicate. Data were expressed as mean ± standard deviation (SD) with statistical significance *p* < 0.05. Statistical analysis was performed using one-way ANOVA through SPSS 22.0.

## 4. Conclusions

In this study, the acidic heteropolysaccharide RLPa-2 was isolated and purified from the *R. laevigata* with Mw of 15.6 kDa, composed of Ara, Gal, Rha, Glc, Xyl, and Gal-UA in a molar ratio of 1.00:0.91:0.39:0.34:0.25:0.20. Structural analysis indicated that RLPa-2 might consist of →6)-α-D-Galp-(1→, →4)-α-D-GalpA-(1→, α-L-Araf-(1→, →2,4)-α-D-Glcp -(1→, β-D-Xylp and α-L-Rhap. Furthermore, RLPa-2 inhibited the production of inflammatory factors (NO, IL-6, and TNF-α), and the expression of pro-inflammatory proteins (p-STAT3/STAT3) in LPS and IFN-γ-induced M1 macrophage, which might be related to the galactose contained in the polysaccharide. Our results demonstrate that RLPa-2 might be a potential resource for anti-inflammatory drug development.

## Figures and Tables

**Figure 1 ijms-25-02133-f001:**
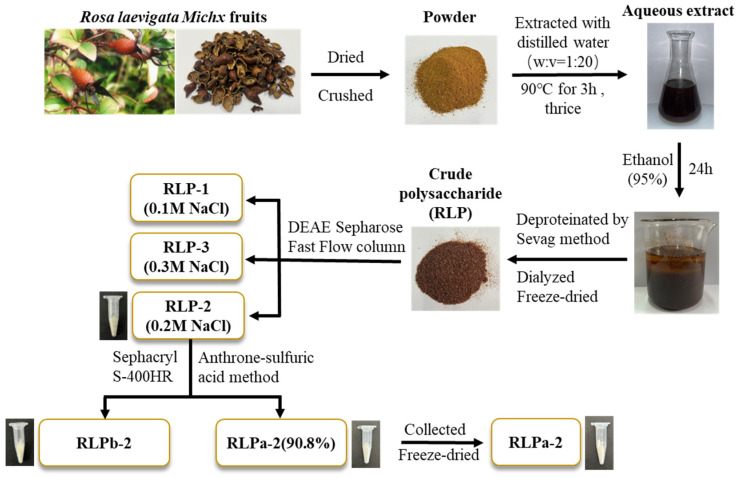
The flowchart illustrating the extraction and purification polysaccharides from *R. laevigata*.

**Figure 2 ijms-25-02133-f002:**
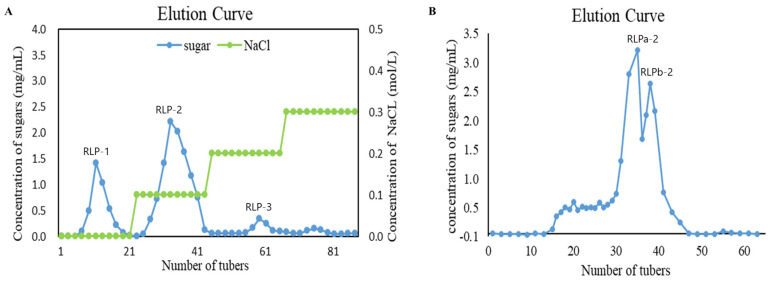
Chemical structure analysis of RLPa−2. (**A**) DEAE Sepharose Fast Flow Chromatogram of RLP. (**B**) Elution profile of RLP−2 on a Sephacryl S−400HR column.

**Figure 3 ijms-25-02133-f003:**
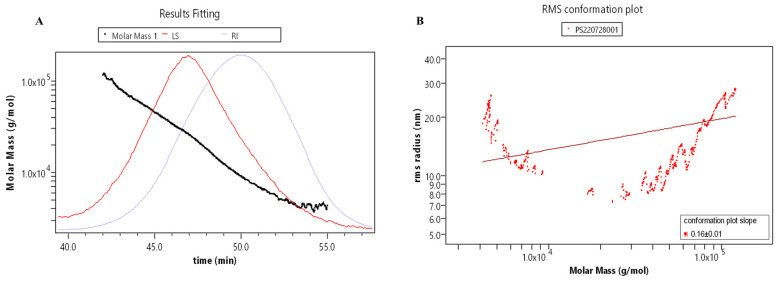
(**A**) Absolute molecular weight analysis graph: the red line represents the multiangle laser light scattering signal (LS); the blue line represents the difference signal (RI); the black line is the molecular weight fitted by the two signals. (**B**) Molecular conformation diagram: log (Molar Mass) is the horizontal coordinate, and log (R.M.S. Radius) is the vertical coordinate, the slope of which can be used as a reference for the molecular conformation.

**Figure 4 ijms-25-02133-f004:**
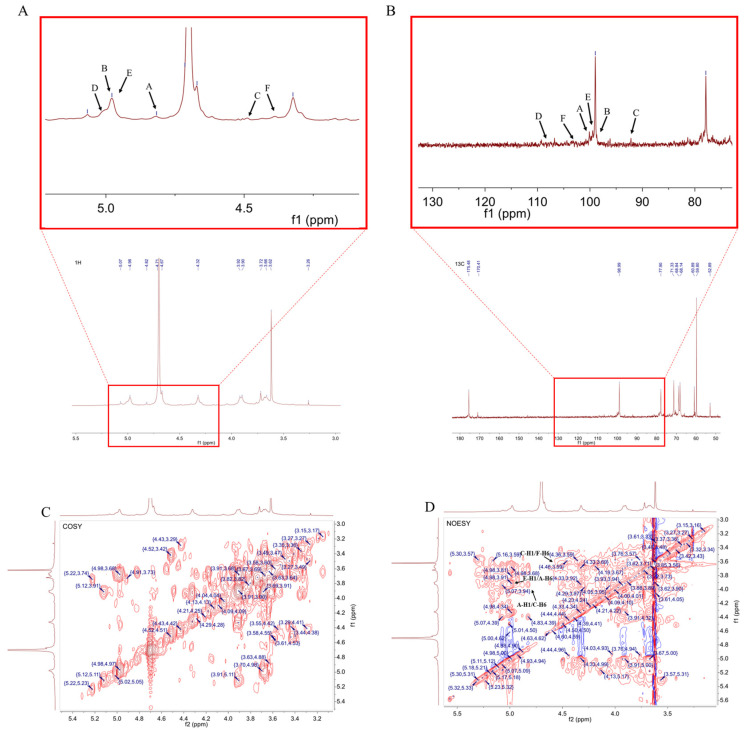
NMR spectrum and presumed principal structure of RLPa−2. (**A**) ^1^H NMR spectrum of RLPa−2; A−F correspond to the H−1 positions of residues A–F, respectively. (**B**) ^13^C NMR spectrum of RLPa−2; A−F correspond to the anomeric carbon signals of residues A–F, respectively. (**C**) ^1^H-^1^H COSY spectrum of RLPa−2. (**D**) ^1^H−^1^H NOESY spectrum of RLPa−2. (**E**) HSQC spectrum of RLPa−2. (**F**) HMBC spectrum of RLPa−2. (**G**) Predicted structure of RLPa−2.

**Figure 5 ijms-25-02133-f005:**
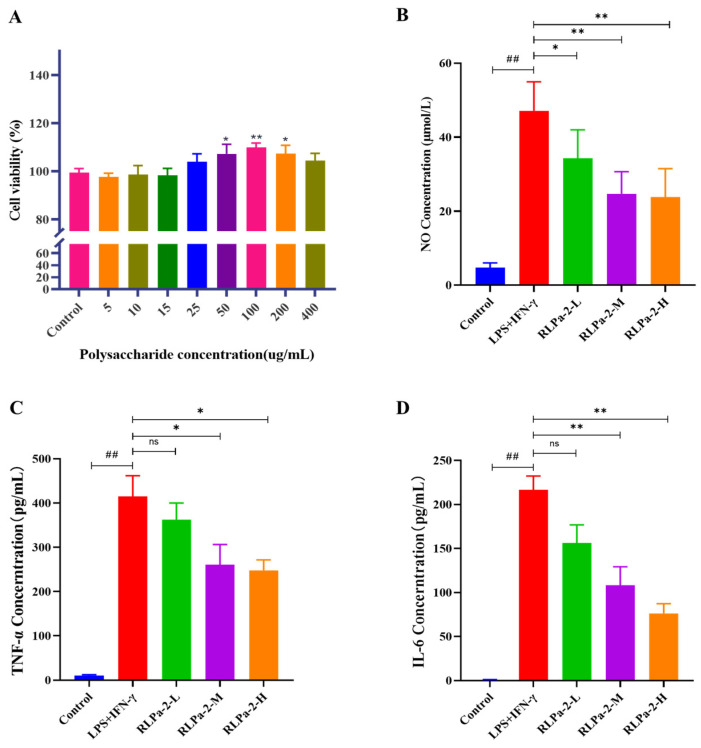
Effect of RLPa−2 on cell viability and pro-inflammatory mediators. (**A**) cell viability (**B**) NO concentration. (**C**) TNF−α concentration. (**D**) IL−6 concentration. Values are presented as mean ± SD (n ≥ 6); Significant difference: * *p* < 0.05, and ** *p* < 0.01 for a difference from the LPS+IFN−γ group, ## *p* < 0.01 as compared with the blank group. “ns” represents no significant difference.

**Figure 6 ijms-25-02133-f006:**
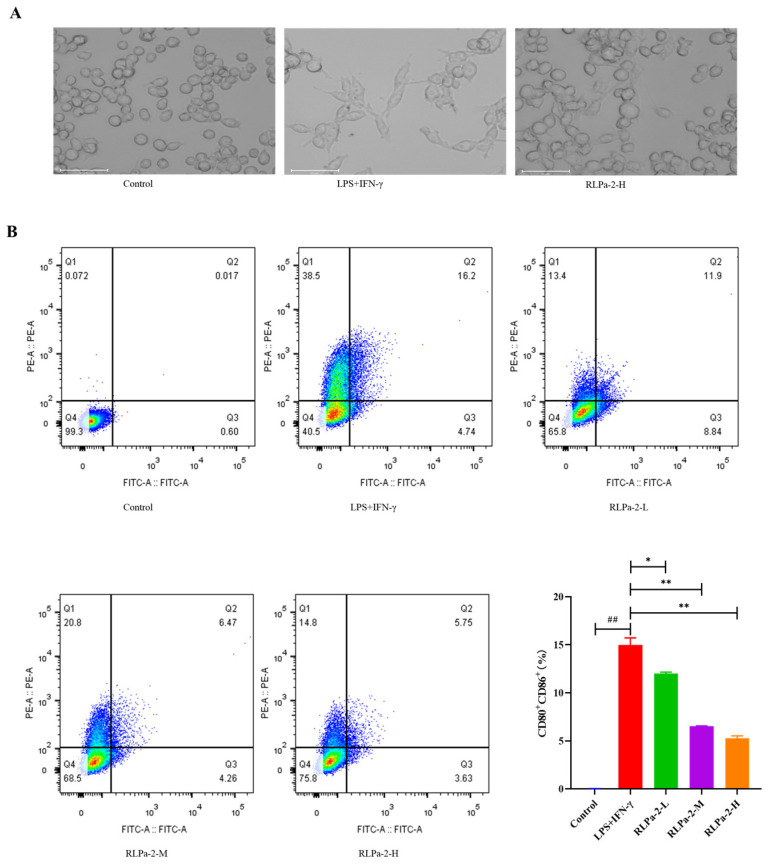
Effect of RLPa−2 on M1 macrophage phenotype and morphology. (**A**) morphological change of macrophages (×100 magnification). The scale bar represents 50 μm. (**B**) The expression of CD80 and CD86 on M1 macrophages. Different colors signify varying cell densities in descending order: red indicates the highest density, followed by yellow, green, and finally, blue represents the lowest. Values are mean ± SD (n ≥ 3); Significant difference: * *p* < 0.05, and ** *p* < 0.01 for the difference from the LPS+IFN−γ group, ## *p* < 0.01 as compared with the blank group.

**Figure 7 ijms-25-02133-f007:**
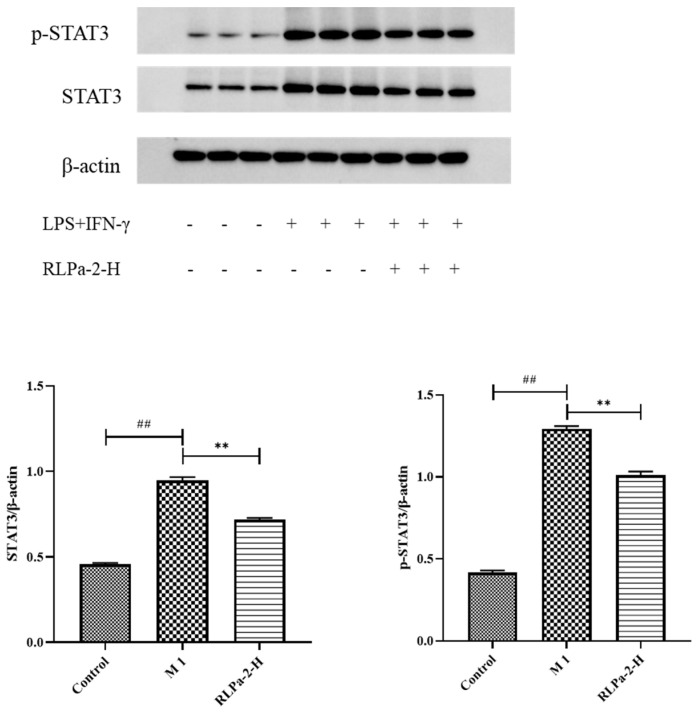
Effect of RLPa−2 on STAT3 protein expression. p−STAT3: STAT3 phosphorylation; STAT3: total STAT3. The protein expression of p−STAT3 and STAT3 was detected and corrected with β−actin to obtain the relative STAT3 and STAT3 phosphorylation measurements among the samples. Data represent mean ± SD from three independent experiments performed in triplicate. Different symbols indicate significant differences according to ANOVA and Tukey test. Significant difference: ** *p* < 0.01 for the difference from the LPS+IFN−γ group, ## *p* < 0.01 compared to the blank group.

**Table 1 ijms-25-02133-t001:** The results of methylation analysis for RLPa−2.

Linkage Type	Partially Methylated Alditol Acetate (PMAAs)	Mass Fragment (*m*/*z*)	Retention Time (min)	Molecular Weight (MW)	Molar Ratio (%)
t-Rha(*p*)	1,5-di-O-acetyl-6-deoxy-2,3,4-tri-O-methyl rhamnitol	72, 89, 102, 118, 131, 162	5.283	293	5.09
t-Ara(*f*)	1,4-di-O-acetyl-2,3,5-tri-O-methyl arabinitol	87, 102, 118, 129, 145, 161	5.535	279	24.62
t-Xyl(*p*)	1,5-di-O-acetyl-2,3,4-tri-O-methyl xylitol	88, 101, 102, 117, 118, 161, 162	6.769	279	7.43
4-Gal(*p*)-UA	1,4,5-tri-O-acetyl-2,3,6-tri-O-methyl galactitol	71, 118, 173, 203, 233	13.028	353	11.63
6-Gal(*p*)	1,5,6-tri-O-acetyl-2,3,4-tri-O-methyl galactitol	87, 99, 102, 118, 129, 162, 189, 233	14.707	351	6.52
2,4-Glc(*p*)	1,2,4,5-tetra-O-acetyl-3,6-di-O-methyl glucitol	71, 88, 113, 130, 173, 190, 211, 233	16.161	379	13.85
4,6-Gal(*p*)	1,4,5,6-tetra-O-acetyl-2,3-di-O-methyl galactitol	71, 85, 118, 159, 201, 261	17.926	379	20.13
3,6-Gal(*p*)	1,3,5,6-tetra-O-acetyl-2,4-di-O-methyl galactitol	87, 101, 118, 129, 189, 234	18.092	379	10.73

**Table 2 ijms-25-02133-t002:** Chemical shifts of sugar residues ^1^H and ^13^C.

Code	Glycosyl Residues	Chemical Shifts (ppm)					
		H1/C1	H2/C2	H3/C3	H4/C4	H5/C5	H6a,6b/C6
A	→6)-*α*-D-Gal*p*-(1→	4.82	3.65	4.01	3.82	3.97	3.92, 3.66
		100.06	68.33	70.04	69.55	74.16	68.89
B	→4)-*α*-D-Gal*p*A-(1→	4.98	3.68	5.06	4.32	4.67	/
		98.94	69.67	70.63	77.94	71.43	175.48
C	→4,6)-*β*-D-Gal*p*-(1→	4.51	3.41	n.d	3.87	3.97	3.91, 3.72
		96.22	71.49	n.d	76.48	74.31	68.81
D	*α*-L-Ara*f*-(1→	5	3.67	n.d	n.d	3.72	/
		107.52	72.33	n.d	n.d	60.9	/
E	→2,4)-*α*-D-Glc*p*-(1→	4.97	3.67	3.35	4.49	n.d	n.d
		99.47	76.13	72.02	79.07	n.d	n.d
F	→3,6)-*β*-D-Gal*p*-(1→	4.43	3.26	4.05	n.d	n.d	3.67, 3.91
		102.8	73.31	83.99	n.d	n.d	68.15

n.d: Abbreviation for “not detected”, which means not recognized.

## Data Availability

The data presented in this study are available in the article.

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
