# Peer review of "Structural Characterization and In Vitro Anti-Inflammatory Activity of Polysaccharides Isolated from the Fruits of Rosa laevigata"

_ijms, 2024, doi:10.3390/ijms25042133_

Round 1
Reviewer 1 Report
Comments and Suggestions for Authors
Manuscript by Peng et al. is devoted to isolation of new polysaccharide from Rosa species and its characterization (structure and composition) by NMR and SEC methods. The authors also showed that polysaccharide obtained has anti-inflammatory effect due to inhibiting the production of inflammatory factors and expression of pro-inflammatory proteins.
The work leaves a pleasant impression. However, to make the study feel complete, I would recommend that the authors add a section on the behavior of the polysaccharide in solution: solubility, tendency of macromolecules to aggregate, charge of particles, etc.
And some minor comments to the MS design:
line 13, “Rosa laevigata Michx” should be italic
lines 30, 47, 86, 87, 88б 295, 305, 306, and 420 “R. laevigata” should be italic
line 37, “Polysaccharides” shouldn’t be capitalized
Figs. 2E and 2F contain captions in Mandarin. Provide it in English
line 211, “in vitro” should be italic
Comments on the Quality of English LanguageMinor editing of English language required
Reviewer 2 Report
Comments and Suggestions for Authors
This work presents a study of the structural and anti-inflammatory features of polysaccharides.
The results and discussion are presented in detail.
However, there are some controversial conclusions and errors.
I urge you to improve the quality of all images in the manuscript.
My main comments concern the quality and validity of the interpretation of data on the structural features of polysaccharides.
It is better to divide Figure 2 into several corresponding figures.
Also, Figs 2E, 2F are probably redundant, and they can be transferred to the saplimentari
Figure 2C is not very informative; it is probably better to present the molecular weight distribution curve in Mw coordinates. Also in section 2.2 there is data on the MV of the polysaccharide, but such accuracy is excessive.
Fig 2D How accurate are the data obtained? According to this figure, the sizes of molecules with MW ~5000 and 100000 (!) are the same and amount to 20 nm.
The conformation determined from the data obtained also raises questions. As is known, polysaccharides are quite flexible in solutions and the usual tilt angle in such coordinates is in the range of 0.6-0.8, which corresponds to mobile polymer coils rather than a “hard sphere” conformation. Additionally, if we conditionally divide the obtained data into 2 ranges - low MW and high MW, then in the range >10000 the slope angle will more accurately characterize the conformation of the polysaccharide.
Based on the above, the authors’ conclusions about the positive relationship between conformational characteristics and anti-inflammatory activity are questioned (line 105-106)
Reviewer 3 Report
Comments and Suggestions for Authors
This is an interesting research on eveluating the anti-inflammatory activity of polysaccharides isolated from the fruits of Rosa laevigata. However, I am wondering if this concept has enough novelty for publication in the Journal. I think it should be revised and clarified several major issues.
1. The author have to inducate that there has been any reported study on the Rosa laevigata extract? It should be mentioned in the introduction. I think that this kind of extract was already studied and published.
2. In the introduction, the authors also need to point out the novelty of this study and its sciencetific contribution
3. The extract method in this study is simply hot water. It is low efficiency. Why did you decide to choose this technique? The other methods like microwave-assisted methods or ultrasonicate-asisted methods should be considered and compared.
4. Figure 1 is very poor. It must be designed more attractively by adding some real images of fruits and extract solutions
5. The authors should compareanti-inflammatory activity of the extract in this study with the similar extracts that have been reported
Comments on the Quality of English LanguageN/A
Round 2
Reviewer 1 Report
Comments and Suggestions for Authors
The authors made required corretions, so MS can be accepted in the current form
Author Response
Thank you once more for your professional review.
Reviewer 2 Report
Comments and Suggestions for Authors
I thank the authors of the manuscript for their answers.
There are a few more questions:
1) Are there any restrictions on the size of polymer particles when using the MALS method?
2) Why is the difference in retention time between the RI detector and the MALS detector so large?
3) Line 349 - which OH-pak HQ column was used third in the system? Which polymer was the DMSO-LiBr system used to analyze?
Reviewer 3 Report
Comments and Suggestions for Authors
it can be accepted
Comments on the Quality of English LanguageN/a
Author Response
Thanks for your professional review. In the revised manuscript, we have implemented minor modifications to the English language, which are highlighted in red for identification (Lines 18, 19, 36, 91, 92, 103, 111, 112, 115, 127, 141, 154, 268, 316, 319, 321, 420, and 424).